# Deubiquitinase USP13 dictates MCL1 stability and sensitivity to BH3 mimetic inhibitors

Shengzhe Zhang[1,2], Meiying Zhang[3], Ying Jing[1], Xia Yin[3], Pengfei Ma[1], Zhenfeng Zhang[4], Xiaojie Wang[5], Wen Di[1,3] & Guanglei Zhuang [1,3]

MCL1 is a pivot member of the anti-apoptotic BCL-2 family proteins. While a distinctive feature of MCL1 resides in its efficient ubiquitination and destruction, the deubiquitinase USP9X has been implicated in the preservation of MCL1 expression by removing the poly-ubiquitin chains. Here we perform an unbiased siRNA screen and identify that the second deubiquitinase, USP13, regulates MCL1 stability in lung and ovarian cancer cells. Mechanistically, USP13 interacts with and stabilizes MCL1 via deubiquitination. As a result, *USP13* depletion using CRISPR/Cas9 nuclease system inhibits tumor growth in xenografted nude mice. We further report that genetic or pharmacological inhibition of USP13 considerably reduces MCL1 protein abundance and significantly increases tumor cell sensitivity to BH3 mimetic inhibitors targeting BCL-2 and BCL-XL. Collectively, we nominate USP13 as a novel deubiquitinase which regulates MCL1 turnover in diverse solid tumors and propose that USP13 may be a potential therapeutic target for the treatment of various malignancies.

[1] State Key Laboratory of Oncogenes and Related Genes, Department of Obstetrics and Gynecology, Ren Ji Hospital, School of Medicine, Shanghai Jiao Tong University, Shanghai 200127, China. [2] School of Biomedical Engineering & Med-X Research Institute, Shanghai Jiao Tong University, Shanghai 200240, China. [3] Shanghai Key Laboratory of Gynecologic Oncology, Ren Ji Hospital, School of Medicine, Shanghai Jiao Tong University, Shanghai 200127, China. [4] State Key Laboratory of Oncogenes and Related Genes, Shanghai Cancer Institute, Ren Ji Hospital, School of Medicine, Shanghai Jiao Tong University, Shanghai 200032, China. [5] Department of Obstetrics and Gynecology, Tongren Hospital, Shanghai Jiao Tong University School of Medicine, Shanghai 200050, China. Shengzhe Zhang, Meiying Zhang and Ying Jing contributed equally to this work. Correspondence and requests for materials should be addressed to G.Z. (email: zhuangguanglei@gmail.com)

Protein ubiquitination is a reversible post-translational modification process that regulates many vital signaling pathways during tumorigenesis[1–3]. Ubiquitination is catalyzed by the concerted actions of E1 activating, E2 conjugating, and E3 ligating enzymes that covalently couple target proteins with ubiquitin and consequently lead to different biological outcomes, especially proteasomal degradation[4, 5]. On the contrary, deubiquitination occurs when deubiquitinases (DUBs) depolymerize and remove ubiquitin adducts from ubiquitylated proteins to reverse the functional effects of ubiquitination[6, 7]. To date, ~100 DUBs in human proteome have been described and classified into seven subfamilies based on the protease domains[8–10], including ubiquitin-specific proteases (USPs), ubiquitin carboxyl-terminal hydrolases (UCHs), Otubain proteases (OTUs), Machado–Joseph disease proteases (MJDs), JAMM/MPN metalloproteases (JAMMs), and the more lately discovered monocyte chemotactic protein-induced proteins (MCPIPs) and motif interacting with Ub-containing novel DUB family (MINDY). In recent years, a plethora of key proteins implicated in oncogenesis, such as p53, PTEN, c-Myc, etc., have been revealed to be exquisitely regulated by one or more deubiquitinating enzymes[11–19]. Therefore, DUBs are emerging as a class of attractive therapeutic targets for cancer, the inhibition of which, under many circumstances, represents an alternative strategy to address the undruggability of their substrates[20]. For example, P5091, a small-molecule inhibitor of USP7, activates HDM2/p53/p21 signaling axis and exerts cytotoxicity in several multiple myeloma (MM) cell models, supporting future clinical investigations of USP7 inhibitors for the treatment of malignant hematological diseases[21].

The B cell lymphoma 2 (BCL-2) family, composed of pro-apoptotic and anti-apoptotic proteins, play a central role in regulating the intrinsic apoptotic pathway. The anti-apoptotic members of the BCL-2 family, including BCL-2, BCL-XL, MCL1 (myeloid cell leukemia sequence 1), BCL-W, A1, and BCL-B, potentiate neoplastic progression and chemotherapy resistance by attenuating cell apoptosis, and are frequently dysregulated in a variety of human cancers[22, 23]. Accordingly, the development of pharmaceutical inhibitors against BCL-2 family proteins as effective anti-cancer therapeutics has been extensively explored[24, 25]. Recent efforts combining nuclear magnetic resonance (NMR)-based screening, fragment chemistry and structure-assisted drug design have resulted in the seminal discovery of ABT-737, a potent BH3 mimetic inhibitor disrupting interactions between anti-apoptotic and pro-apoptotic BCL-2 proteins[26]. Subsequently, the orally bioavailable analog ABT-263 (navitoclax) was evaluated in clinical trials and delivered favorable antitumor activity despite dose-limiting thrombocytopenia associated with BCL-XL inhibition[27]. ABT-199 (venetoclax), a highly selective BCL-2 inhibitor that spares platelets, was then designed and approved by the Food and Drug Administration (FDA) for patients with chronic lymphocytic leukemia (CLL) harboring 17p deletion who have received at least one prior treatment[28]. However, all current BCL-2 family inhibitors cannot engage the more divergent MCL1 molecule, which greatly constrains the cytotoxic action of BH3 mimetic compounds[29, 30], and the generation of high-affinity inhibitors directly targeting MCL1 remains challenging[31]. MCL1 is unique due to its short protein half-life and previous studies have elucidated that multiple E3 ubiquitin ligases, such as MULE, SCF[Fbw7] and APC/C[Cdc20], efficiently polyubiquitylate MCL1 for degradation[32–35]. Inversely, deubiquitinase USP9X stabilizes MCL1 by removing the polyubiquitin chains, and thus has been considered as a potential prognostic and therapeutic target in several human malignancies[36]. Nevertheless, USP9X exhibits tissue-specific expression primarily in brain and the immune system[37], and occasionally

acts as a tumor suppressor, e.g., in oncogenic KRAS-initiated pancreatic carcinoma[38], suggesting the possible existence of additional DUBs that regulate MCL1 stability.

In this study, we find that in certain lung and ovarian cancer cell lines, USP9X knockdown does not alter MCL1 protein levels. We perform a human deubiquitinase short interfering RNA (siRNA) library screen and identify that USP13 (ubiquitin-specific protease 13) functions as a novel MCL1 DUB to enhance its stability and promote tumor survival. USP13 and MCL1 display increased copy numbers in many TCGA (The Cancer Genome Atlas) cancer types, and are correlatively upregulated only at protein level in lung and ovarian tumor specimens. In addition, genetic depletion of USP13 using clustered regularly interspaced palindromic repeats (CRISPR)/Cas9, or pharmacological inhibition of USP13 by a small-molecule inhibitor spautin-1, markedly downregulates MCL1 protein expression and shows synergistic effects against tumor cells in combination with ABT-263, a selective antagonist of BCL-2 and BCL-XL. Therefore, we propose that deubiquitinase USP13 is a new regulator of MCL1 stability and drug sensitivity to BH3 mimetic inhibitors, and may represent a promising therapeutic target for cancer treatment.

## Results

**Identification of USP13 as a candidate MCL1 deubiquitinase**. To systematically identify DUBs that may regulate MCL1 stability, we performed a deubiquitinase knockdown screen in HEK293T cells and focused on 20 candidate DUBs whose knockdown led to reduced MCL1 abundance (Fig. 1a). The siRNA library consisted of pooled oligos (a mixture of four siRNAs per DUB) targeting each of 84 human DUBs spanning five subfamilies (Fig. 1b and Supplementary Data 1). The MCL1 protein levels were determined by western blot analysis (Supplementary Fig. 1a). In parallel, given the important role of MCL1 in neoplasm, we conducted TCGA pan-cancer genomic interrogation of copy number alterations (Supplementary Fig. 1b and Supplementary Table 1) and somatic mutations (Supplementary Fig. 1c and Supplementary Table 2) for the tested 84 human DUBs. Among the 20 candidate MCL1-regulating DUBs, USP13 drew our attention because it was amplified in several types of human cancer (Fig. 1c), particularly in 45.9% of lung squamous cell carcinoma (504 samples) and 28% of ovarian serous adenocarcinoma (603 samples). We confirmed the siRNA screen results by transfecting each of the four oligos against USP13 into HEK293T cells, and indeed found that USP13 knockdown decreased the endogenous protein levels of MCL1, while modulating USP9X or OTUB2 with different siRNAs had no consistent effects on MCL1 expression (Fig. 1d). Unlike previous reports[13], we did not observe altered PTEN levels upon USP13 downregulation in our model. Taken together, these data suggested that USP13 might be a novel candidate deubiquitinase for MCL1.

**USP13 and MCL1 are correlatively upregulated in cancer**. We sought to further investigate cancer-associated alterations of USP13 and MCL1 in more detail and determine the clinical relevance of USP13-MCL1 regulatory axis. Copy number analysis of TCGA tumor samples in cBioportal database showed that both USP13 and MCL1 were genomically amplified in a wide range of cancer types (Fig. 2a), in particular a considerable proportion of lung adenocarcinoma, lung squamous cell carcinoma and ovarian serous carcinoma cases (Fig. 2b). In contrast, we did not observe copy number changes of USP9X in the majority of malignancies including lung and ovarian cancer (Supplementary Fig. 2a). Consistently, western blot analysis of 22 lung cancer and 33

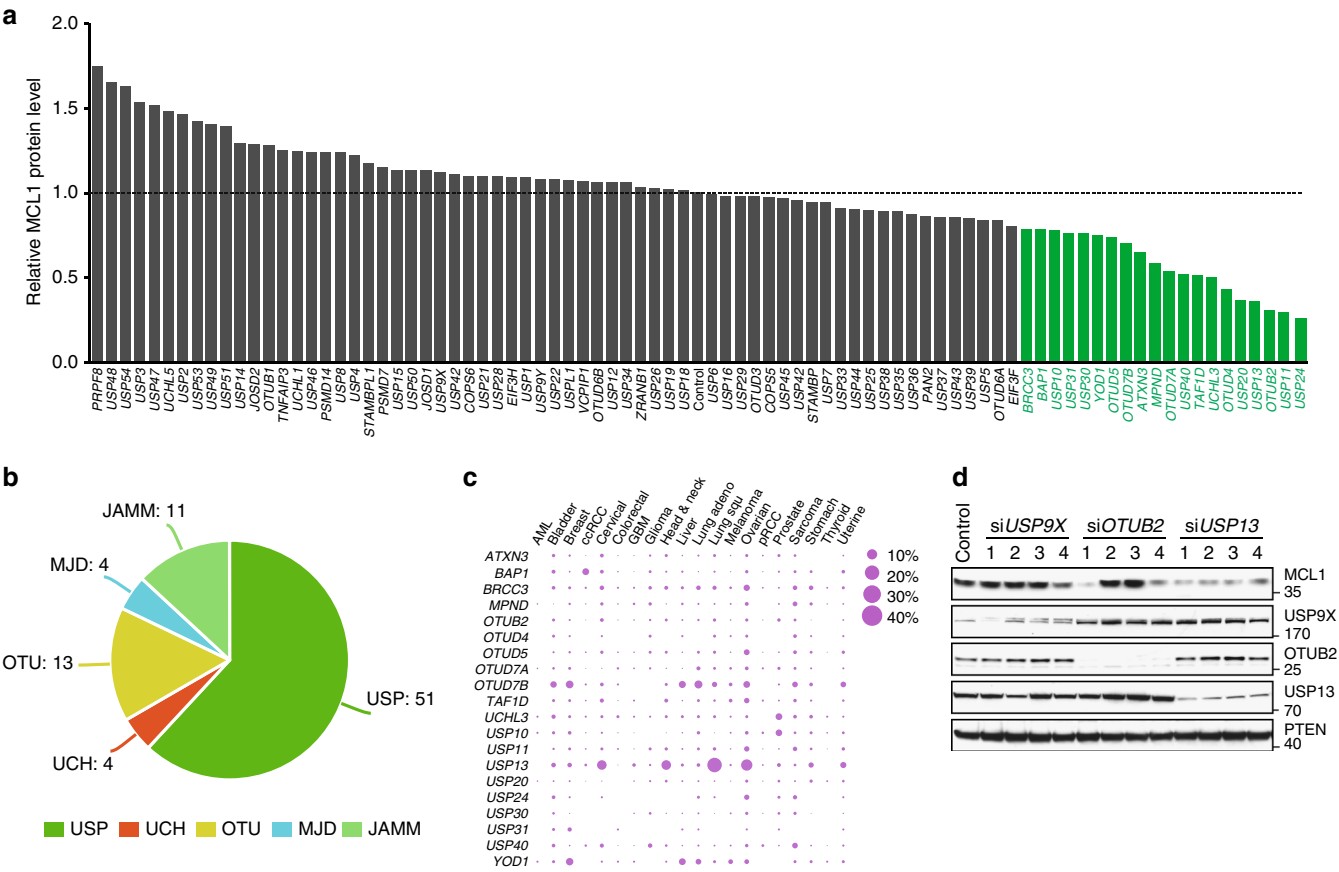

**Fig. 1** Identification of USP13 as a candidate MCL1 deubiquitinase. **a** The quantification of MCL1 protein levels upon knockdown of each DUB in HEK293T cells. 84 siRNA pools were individually transfected into HEK293T cells, endogenous MCL1 was detected by western blotting and protein bands were quantified using the Image J software. **b** Classification of the 84 human DUBs included in siRNA screen. **c** Dot plot of copy number alterations for 20 candidate MCL DUB genes in TCGA cancers. **d** Validation of the candidate MCL1 DUBs in HEK293T cells. Four different siRNAs targeting USP9X, OTUB2, or USP13 were transfected into HEK293T cells, and endogenous expression levels of indicated proteins were measured by western blot analysis

ovarian cancer cell lines revealed that both USP13 and MCL1, but not USP9X, were ubiquitously expressed at the protein level (Fig. 2c). Notably, gene expression of USP13 and MCL1 correlated with the genomic amplification status (Supplementary Fig. 2b), indicative of their functional importance in driving lung and ovarian cancer. However, correlation analysis of RNA sequencing data from TCGA failed to provide evidence that USP13 and MCL1 were associated with each other at the messenger RNA (mRNA) level (Supplementary Fig. 2c), implying that USP13-mediated regulation of MCL1, if any, might not occur through a transcriptional mechanism. To determine the protein expression and relatedness of USP13 and MCL1 in patients, we performed immunohistochemical (IHC) staining of human lung and ovarian cancer tissues. USP13 and MCL1 were readily detectable in most ovarian cancer specimens (Supplementary Table 3), and analysis of consecutive tissue sections discovered a significantly positive correlation between USP13 and MCL1 expression (Spearman's rank correlation coefficient R = 0.60, P = $2.61 \times 10^{-4}$) (Fig. 2d, e). Similarly, IHC assessment of a lung cancer tissue microarray (Supplementary Table 4) showed that USP13 and MCL1 were upregulated in cancers compared with the matched adjacent normal tissues, and their expression tended to correlate (Spearman's rank correlation coefficient R = 0.46, P = 0.13). Specifically, USP13 and MCL1 expression was both observed in 93.3% (28 of 30) of our lung cancer cohort, whereas only 26.7% (8 of 30) and 3.3% (1 of 30) of normal lung tissues displayed positive staining of USP13 and MCL1, respectively (Fig. 2d, e). Collectively, these findings supported that USP13 and

MCL1 were concomitantly overexpressed in human lung and ovarian cancer to promote tumorigenesis.

**USP13 promotes MCL1 protein stability.** The fact that the correlation between USP13 and MCL1 expression was restricted to the protein level prompted us to hypothesize that USP13 could regulate MCL1 as a deubiquitinase enzyme in lung and ovarian cancer. To confirm whether USP13 dictated the stability of MCL1 protein in tumor cells, we introduced two independent siRNAs against USP13 into several lung cancer (SW-1573, NCI-H441 and A549) and ovarian cancer (TOV-21G, HEY and OVCA433) cell lines (Fig. 3a). As expected, USP13 knockdown resulted in uniform downregulation of endogenous MCL1 protein (Fig. 3a) without notable effect on MCL1 mRNA expression (Fig. 3b), whereas PTEN or other BCL-2 family members were not conclusively affected (Supplementary Fig. 3a). SW-1573 and TOV-21G cells treated with the proteasome inhibitor MG132 contained higher levels of MCL1 protein (Supplementary Fig. 3b). We also tested siRNAs targeting USP9X in these cell models and did not detect a resultant decrease of MCL1 protein (Supplementary Fig. 3c), arguing that USP13 instead of USP9X served as a regulator of MCL1 stability. HUWE1 (encoding MULE) and FBXW7 (encoding SCF^Fbw7), two putative E3 ubiquitin ligases of MCL1, was knocked down individually or combinatorially in SW-1573 and HEY cells. In all conditions, MCL1 protein levels were markedly increased, and the protective effects of USP13 were diminished only when both MULE and SCF^Fbw7 were absent

(Supplementary Fig. 3d). As a complimentary approach, we lentivirally expressed ectopic *USP13* into OVCAR5, PC9 and NCI-H2170 cells, whose endogenous USP13 levels were relatively low (Fig. 3c), and found that overexpression of *USP13* increased MCL1 protein, but not mRNA, abundance (Fig. 3c, d). Next, we

examined the impact of USP13 on MCL1 protein turnover in the presence of cycloheximide (CHX), an inhibitor of protein translation. *USP13* knockdown by siRNA markedly reduced the half-life of MCL1 in TOV-21G (Fig. 3e) and SW-1573 cells (Fig. 3f). Conversely, exogenous *USP13* overexpression evidently

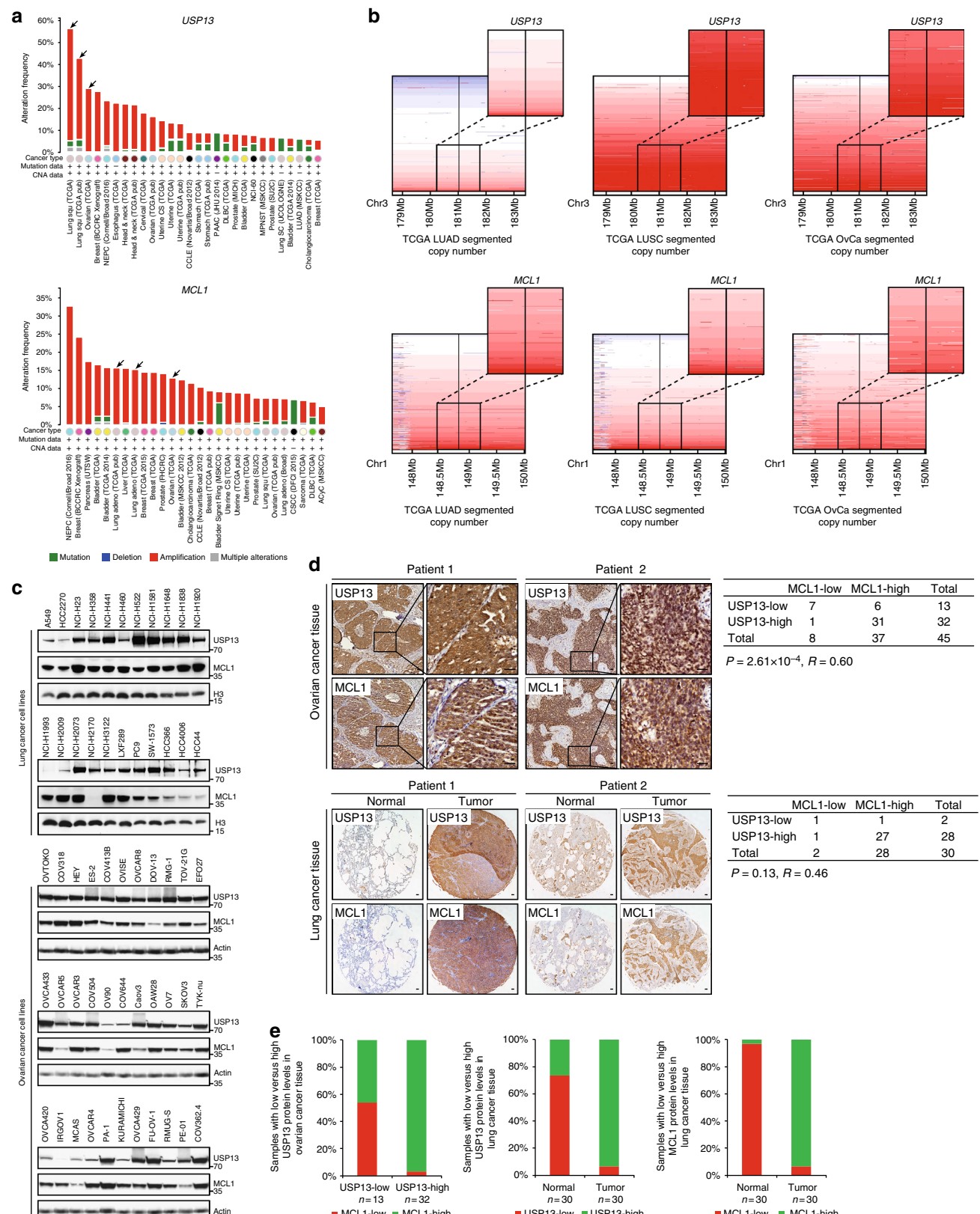

prolonged the half-life of endogenous MCL1 in NCI-H2170 (Fig. 3g) and PC9 cells (Fig. 3h). We concluded that USP13 promoted MCL1 stability at the post transcriptional level in lung and ovarian cancer cells.

**USP13 interacts with and deubiquitinates MCL1.** To gain insights into USP13-mediated regulation of MCL1, we tested whether USP13 interacted with MCL1 and functioned as a deubiquitinase. Protein co-immunoprecipitation analysis was performed with 3×FLAG-tagged USP13 and HA-tagged MCL1 co-transfected into HEK293T cells. Ectopic MCL1 could be detected in USP13 immunoprecipitates using anti-FLAG antibodies (Fig. 4a). Moreover, the interaction of USP13 and MCL1 occurred between endogenous proteins in A549, HEY and SW-1573 tumor cells (Fig. 4b). In addition, cellular fractionation assays showed that a portion of endogenous USP13 was in the membrane-enriched fraction containing most MCL1, and both proteins also resided in the cytosolic compartment (Supplementary Fig. 4a). Immunofluorescence microscopy of HEY cells confirmed that USP13 co-localized with MCL1 (Supplementary Fig. 4b). We further conducted the in vitro immunoprecipitation experiment, the result of which suggested that the interaction between USP13 and MCL1 was direct and independent on MCL1 ubiquitination status (Fig. 4c). USP13 consists of an UBP (ubiquitin-specific processing protease)-type zinc finger, an USP and two UBA (Ubiquitin-associated/translation elongation factor EF1B, amino-terminal) domains; MCL1 contains two PEST (proline gluatamic acid, serine and threonine), four BH (BCL-2 homology) and a carboxyl-terminal transmembrane domains. To map the binding region on USP13 and MCL1, we co-expressed FLAG-tagged USP13 with two deletion mutants of HA-tagged MCL1, or FLAG-tagged MCL1 with two deletion mutants of HA-tagged USP13 (Fig. 4d). Co-immunoprecipitation experiments demonstrated that the amino terminuses of USP13 and MCL1 were essential for the physical interaction of two proteins (Fig. 4e). Finally, we determined whether MCL1 was a deubiquitination substrate of USP13 by transfecting expression vectors encoding MCL1, HA-tagged ubiquitin and wild type or mutant USP13-C345A into HEK293T cells. Cells were treated with MG132, MCL1 protein was purified with anti-MCL1 resins and MCL1 ubiquitination was analyzed using an antibody against HA. Consistent with USP13 removing ubiquitin from MCL1, wild type USP13, but not catalytically inactive mutant USP13-C345A, markedly reduced the amount of ubiquitinated MCL1 (Fig. 4f). Importantly, MCL1 was a direct substrate of USP13, as indicated by the in vitro ubiquitination assay (Fig. 4g). Therefore, USP13 interacted with and deubiquitinated MCL1 to preserve its protein stability.

*USP13 genetic deletion suppresses tumor growth.* We subsequently investigated the impact of USP13-mediated MCL1 stabilization on tumor malignancy. *USP13* was genetically knocked out in TOV-21G and SW-1573 cells using CRISPR/Cas9 system (Fig. 5a). Interestingly, although *USP13* depletion reduced MCL1 protein levels (Fig. 5a), neither cell proliferation (Supplementary Fig. 5a) nor cell migration (Supplementary Fig. 5b) was significantly changed under normal growth conditions. Moreover, *USP13* knockout or knockdown did not seem to disrupt the mitochondrial homeostasis (Supplementary Fig. 5c), in which an amino-terminally truncated isoform of MCL1 has been implicated[39]. However, in the context of hypoxic or oxidative stresses, *USP13* loss caused a substantial decrease of cell viability (Fig. 5b) and a significant increase of cell apoptosis (Fig. 5c) in both TOV-21G and SW-1573 tumor models. We reasoned that MCL1 downregulation due to USP13 elimination might not directly initiate the apoptotic process, but enhance the effects of death signals elicited by unfavorable growth environment or drug treatment. Indeed, when we subcutaneously implanted TOV-21G and SW-1573 cells into nude mice, *USP13* knockout dramatically retarded tumor growth in xenograft models. Specifically, we observed a fivefold decrease in the tumor volume of *USP13*-depleted SW-1573 cells (Fig. 5d) and a complete eradication of tumors formed by *USP13*-null TOV-21G cells (Fig. 5e). Importantly, a partial rescue of xenograft growth was obtained when exogenous MCL1 was expressed in the USP13-deficient SW-1573 cells (Fig. 5f and Supplementary Fig. 5d). These results demonstrated that USP13 promoted tumor development in vivo, which might be, at least in part, through the stabilization of MCL1.

**Targeting USP13 sensitizes tumor cells to BH3 mimetics.** MCL1 has been reported to limit tumor responses to chemotherapeutics and BH3 mimetic inhibitors. To further evaluate its impact on drug sensitivity in ovarian cancer cells, we screened a collection of 180 small-molecule inhibitors (Supplementary Data 2) targeting wide-ranging signaling pathways using *MCL1*-depleted SW-1573 cells (Supplementary Fig. 6a). Consistent with previous reports[29, 30], we found that ABT-263 was among cytotoxic agents which inhibited cell growth of *MCL1*-null SW-1573 to a greater extent compared with parental control (Supplementary Fig. 6b). ABT-263 is a small-molecule antagonist that inhibits the anti-apoptotic activity of BCL-2 and BCL-XL, but not MCL1[27]; hence, tumors expressing abundant MCL1 proteins are expected to resist ABT-263 treatment by compromising apoptosis induction. Therefore, we speculated that USP13 might regulate ABT-263 responsiveness through downregulating MCL1 expression. Because drug development specifically against MCL1 has been considerably slower, targeting USP13 to inhibit MCL1 indirectly is likely to represent an alternative approach to potentiate the therapeutic efficacy of ABT-263 in cancer cells. As anticipated, *USP13* knockdown by siRNAs (Supplementary Fig. 6c) or knockout by CRISPR/Cas9 (Fig. 6a) increased sensitivity to ABT-263 in SW-1573 and TOV-21G cells. Conversely, *USP13* overexpression in NCI-H2170 cells modestly impaired the inhibitory effects of ABT-263 (Supplementary Fig. 6d). These data suggested that genetic inactivation of USP13 could improve tumor cell susceptibility to BH3 mimetic inhibitors such as ABT-263.

---

**Fig. 2** *USP13* and *MCL1* are correlatively upregulated in cancer. **a** Pan-cancer analysis of *USP13* and *MCL1* genomic alterations in cBioportal database. **b** Copy number analysis of *USP13* and *MCL1* in TCGA lung adenocarcinoma, lung squamous cell carcinoma and ovarian serous carcinoma samples. Color scale: amplification in red and deletion in blue. **c** Expression of USP13 and MCL1 in human lung cancer and ovarian cancer cell lines. We used 22 lung cancer and 33 ovarian cancer cell lines to determine the endogenous protein levels of USP13 and MCL1. **d** Representative images for immunohistochemical staining of USP13 and MCL1 protein in human ovarian cancer samples, normal lung and lung cancer tissues (scale bar=50 μm). Statistical significance of correlation between USP13 and MCL1 protein levels was determined by a $\chi^2$ (chi-squared) test. R represented the correlation coefficients. **e** Positive correlation of USP13 with MCL1 protein expression in human ovarian cancer tissues, and enrichment of USP13 or MCL1 expression in human lung cancer specimens relative to normal lung tissues. Forty-five human ovarian cancer samples were categorized into two groups (13 USP13-low and 32 USP13-high), and MCL1 expression status was indicated in each group. Thirty human normal lung or lung cancer tissues were classified based on the USP13 or MCL protein expression status

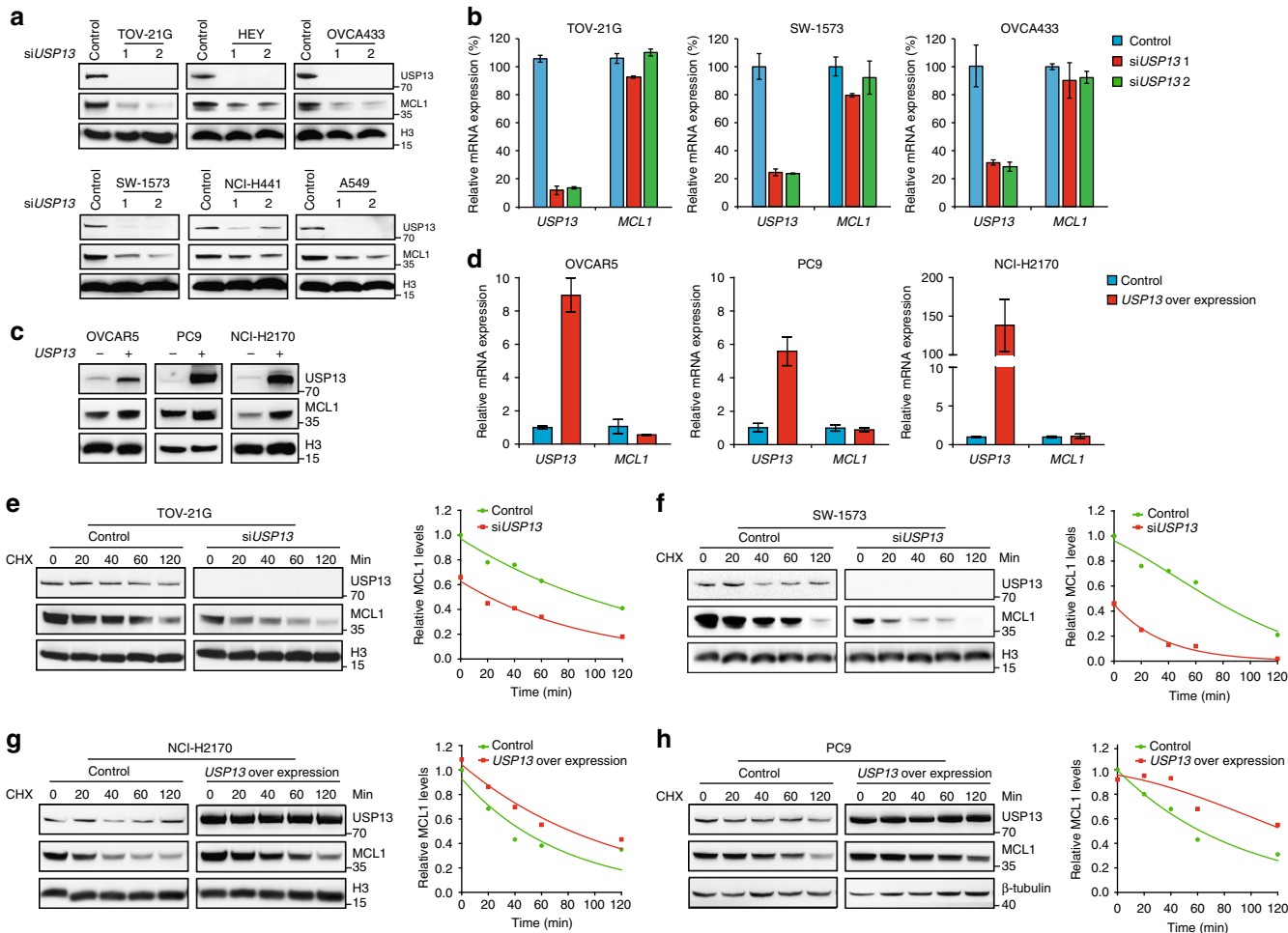

**Fig. 3** USP13 promotes MCL1 protein stability. **a** *USP13* knockdown by siRNAs decreased MCL1 protein levels in human ovarian cancer (TOV-21G, HEY and OVCA433) and lung cancer (SW-1573, NCI-H441 and A549) cell lines. **b** Quantitative PCR analysis of *USP13* and *MCL1* mRNA levels in TOV-21G, SW-1573 and OVCA433 cells upon *USP13* knockdown (*P < 0.05, Student's *t*-test). The gene expression levels were normalized to *GAPDH*, and error bars indicated standard deviation (each condition contained three biological replicates). **c** Lentiviral *USP13* overexpression increased MCL1 protein levels in human ovarian cancer (OVCAR5) and lung cancer (PC9 and NCI-H2170) cell lines. **d** Quantitative PCR analysis of *USP13* and *MCL1* mRNA levels in OVCAR5, PC9 and NCI-H2170 cells upon *USP13* overexpression (*P < 0.05, Student's *t*-test). The gene expression levels were normalized to *GAPDH*, and error bars indicated standard deviation (each condition contained three biological replicates). **e**, **f** Left: TOV-21G or SW-1573 cells were transfected with *USP13* siRNAs, treated with 20 µg/mL CHX, collected at different time points and then immunoblotted with antibodies against indicated proteins. Right: quantification of MCL1 protein levels (normalized to H3) at different time points. **g**, **h** Left: PC9 and NCI-H2170 cells were transduced with lentiviral vector overexpressing *USP13*, treated with 20 µg/mL CHX, collected at different time points and then immunoblotted with antibodies against indicated proteins. Right: quantification of MCL1 protein levels (normalized to H3 or β-tubulin) at different time points

In addition, we explored whether pharmacological inhibition of USP13 downregulated MCL1 protein expression and synergistically kill tumor cells in combination with ABT-263. Spautin-1 (specific and potent autophagy inhibitor-1) was previously discovered as a potent small-molecule inhibitor of USP13 and USP10 [40]. We treated SW-1573 and TOV-21G cells with spautin-1 and found that spautin-1 reduced the protein level of MCL1 (Fig. 6b). The proteasome inhibitor MG-132 completely abrogated the reduction of MCL1 induced by spautin-1 treatment, suggesting that spautin-1 promoted the degradation of MCL1 through the proteasomal pathway. Notably, endogenous MCL1 progressively decreased upon spautin-1 exposure in a dose- and time-dependent manner (Fig. 6c). To determine whether spautin-1 as a USP13 inhibitor could sensitize tumor cells to ABT-263, we performed dose matrix experiments containing a range of drug concentrations and analyzed the effects of combining ABT-263 and spautin-1 using the Bliss independence model[41]. The heatmaps for Bliss synergy scores showed that ABT-263 in combination with spautin-1 synergistically inhibited tumor cell viability across multiple concentrations (Fig. 6d). We validated these findings using crystal violet staining of drug-treated SW-1573 and TOV-21G cells (Fig. 6e). Taken together, our results indicated that pharmacological inhibition of USP13 by spautin-1 reduced MCL1 protein abundance and increased tumor cell sensitivity to ABT-263.

## Discussion

In this study, through an unbiased DUBs-focused siRNA screen, we have identified USP13 as a novel MCL1 deubiquitinase that interacts with MCL1, reverses MCL1 ubiquitylation, protects MCL1 from proteasomal degradation and promotes tumor cell survival, particularly in the presence of extrinsic stresses, such as those generated by hypoxia, oxidants or pro-apoptotic BH3 mimetic inhibitors. Similar to MCL1, USP13 is genomically amplified in a variety of human cancers and ubiquitously

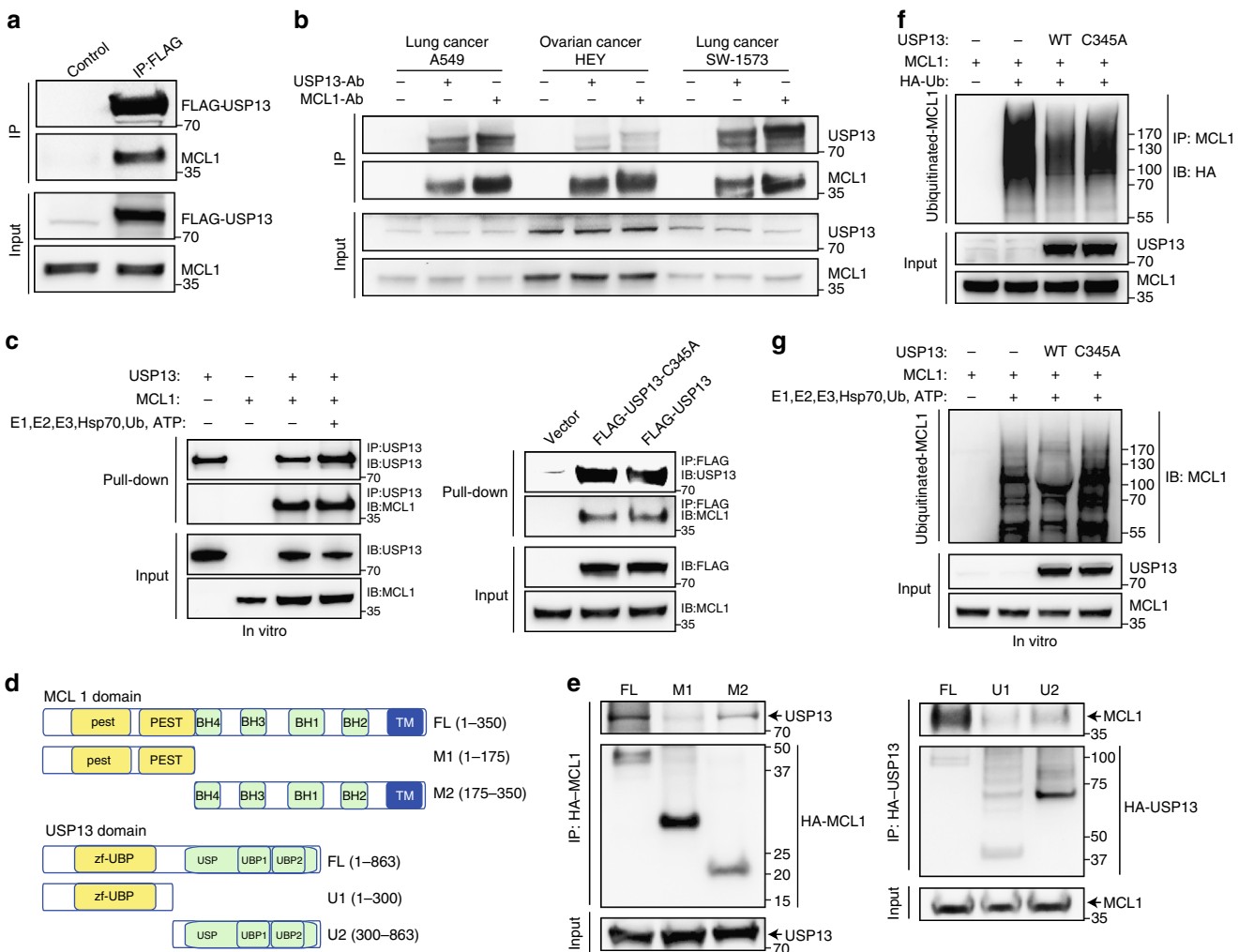

**Fig. 4** USP13 interacts with and deubiquitinates MCL1. **a** 3 × FLAG-tagged USP13 was transfected into HEK293T cells. Cells were treated with MG-132 for 8 h and USP13 was immunoprecipitated using an anti-FLAG antibody. Co-immunoprecipitated MCL1 was detected using an anti-MCL1 antibody. **b** Endogenous USP13 and MCL1 were immunoprecipitated from A549, HEY and SW-1573 cells using an anti-USP13 antibody or an anti-MCL1 antibody respectively. Co-immunoprecipitated endogenous MCL1 or USP13 was analyzed by western blotting. **c** Left: Purified His-MCL1 and His-USP13 proteins were co-incubated and immunoprecipitated with an anti-USP13 antibody. Co-immunoprecipitated MCL1 was detected using an anti-MCL1 antibody. Right: FLAG-USP13 or FLAG-USP13-C345A was transfected into HEK293T cells and immunoprecipitated using an anti-FLAG antibody. Co-immunoprecipitated MCL1 was detected using an anti-MCL1 antibody. **d** A schematic of full-length USP13 (FL), full-length MCL1 (FL) and their deletion mutants (M1, M2, U1, U2). **e** USP13 was co-transfected with the MCL1 deletion mutants, and MCL1 was co-transfected with the USP13 deletion mutants. Interactions were analyzed using the co-immunoprecipitation assay. **f** MCL1 was co-transfected with FLAG-USP13 or the FLAG-USP13-C345A mutant, with or without HA-ubiquitin. The ubiquitinated MCL1 was measured using an in vivo ubiquitination assay. **g** Purified His-MCL1 and His-USP13 proteins were co-incubated in the ubiquitination system. The ubiquitinated MCL1 was measured using an in vitro ubiquitination assay

overexpressed in lung and ovarian tumor specimens. Therefore, our data demonstrate that USP13 is a novel regulator of MCL1 stability and a potential therapeutic target for cancer treatment.

MCL1 is a key member of the anti-apoptotic BCL-2 family and, as an oncoprotein, is frequently dysregulated in many cancers[42]. Despite tremendous efforts, the clinical development of small molecules targeting MCL1 has been challenging. Alternative strategies include chemical compounds that broadly inhibit gene transcription or protein translation and consequently down-regulate MCL1[43]. In addition, MCL1 displays a uniquely rapid turnover rate and depends on deubiquitinases to counteract proteasomal degradation. To our best knowledge, the only reported deubiquitinase that stabilizes MCL1 protein is USP9X, which serves as a promising prognostic and therapeutic target across a range of human malignancies[36]. However, USP9X

exhibits tissue-specific expression and effects[37], and counter-intuitively functions as a tumor suppressor in pancreatic ductal adenocarcinoma[38], raising the likelihood that other DUBs might regulate MCL1 stability under certain circumstances. Indeed, we found that USP9X knockdown did not convincingly alter MCL1 protein levels in multiple lung and ovarian cancer cell lines. Instead, USP13 inhibition considerably reduced MCL1 half-life and protein abundance. Therefore, we propose that USP13 is a bona fide deubiquitinase for MCL1, a notion that is further supported by the endogenous interaction and correlative expression of USP13 and MCL1 only at protein levels in clinical tumor samples. Notably, these findings do not exclude the possibility that additional DUBs other than USP9X or USP13 exist to mediate MCL1 deubiquitination, considering that multiple E3 ubiquitin ligases, such as MULE, SCF[Fbw7] and APC/C[Cdc20], are able to mark MCL1 for degradation[32–35]. It remains to be

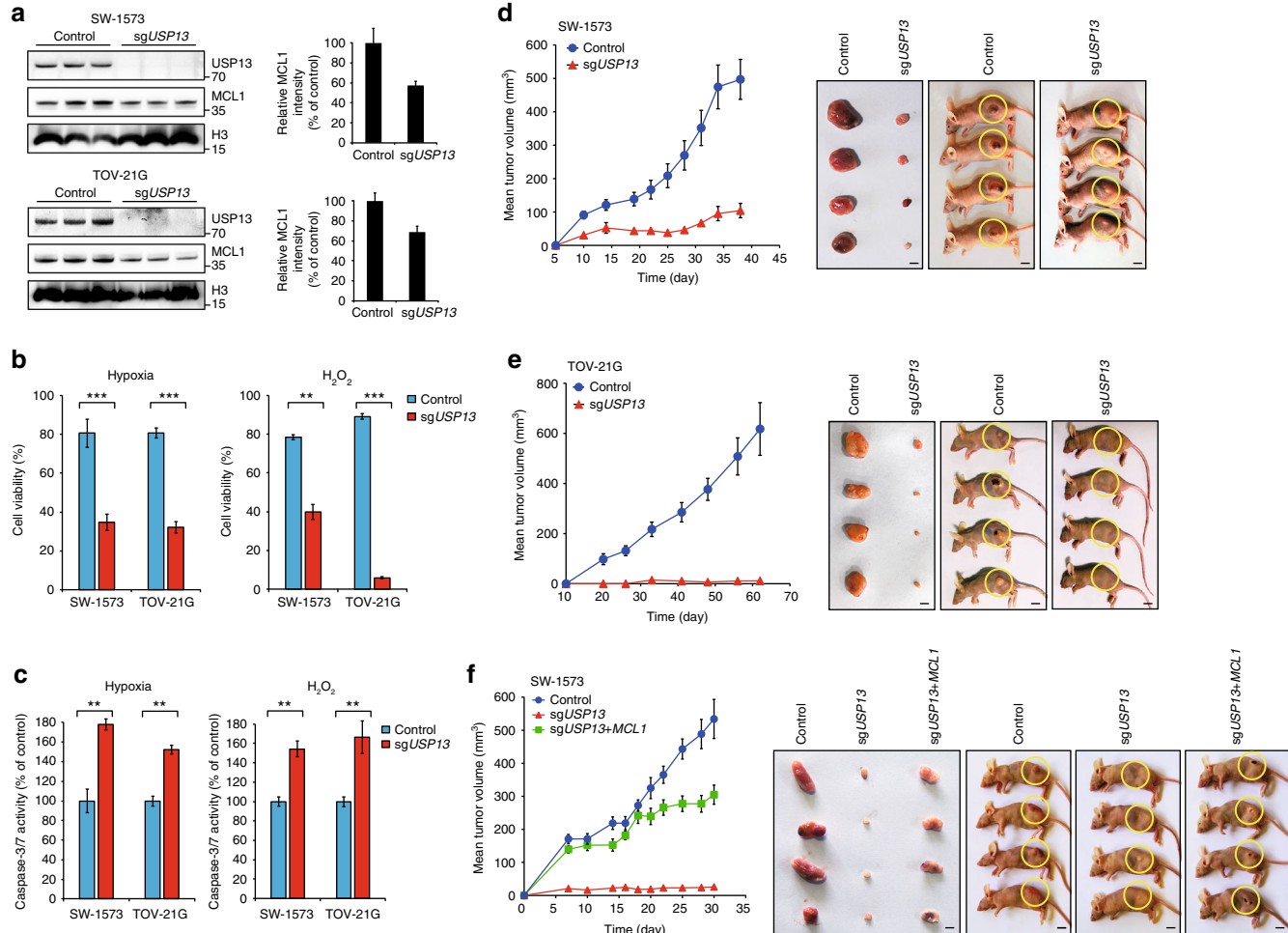

**Fig. 5** *USP13* genetic deletion suppresses tumor growth. **a** *USP13* depletion in SW-1573 and TOV-21G cells using the CRISPR/Cas9 system. The endogenous USP13 and MCL1 protein levels were measured by western blotting. **b** Cell viability upon *USP13* depletion was measured using Cell Counting Kit-8 under hypoxic or oxidative stresses, and error bars indicated standard deviation (each condition contained three biological replicates). **c** Cell apoptosis upon *USP13* depletion was measured using Caspase-Glo 3/7 Assay under hypoxic or oxidative stresses, and error bars indicated standard deviation (each condition contained three biological replicates). **d** Tumor growth curves and representative tumor images were shown for *USP13*-depleted SW-1573 cells subcutaneously implanted into nude mice (scale bar, 10 mm). Error bars indicated standard deviation (10 mice per group). **e** Tumor growth curves and representative tumor images were shown for *USP13*-depleted TOV-21G cells subcutaneously implanted into nude mice (scale bar=10 mm). Error bars indicated standard deviation (10 mice per group). **f** Tumor growth curves and representative tumor images were shown for *USP13*-depleted/*MCL1*-overexpressed SW-1573 cells subcutaneously implanted into nude mice (scale bar, 10 mm). Error bars indicated standard deviation (10 mice per group)

elucidated whether the ubiquitination and deubiquitination regulation of MCL1, dependent on the cell context, may be achieved by involving different pairs of E3 ubiquitin ligase and deubiquitinase.

Previous studies have implicated USP13 deubiquitinase in cancer development by regulating protein stability of oncogenes or tumor suppressors. However, the pathological function of USP13 in tumourigenesis has been controversial in different cancers. For example, USP13 deubiquitinates and stabilizes microphthalmia-associated transcription factor (MITF), a lineage-specific master regulator of human melanoma[44]. Additionally, USP13 plays an oncogenic role in glioblastoma and ovarian cancer by stabilization of c-MYC and ACLY/OGDH, respectively[45, 46]. In contrast, USP13 was discovered as a PTEN deubiquitinase and a tumor-suppressing protein in breast cancer[13]. Here, we provided several lines of evidence to support that MCL1 was a deubiquitination target of USP13 in lung and ovarian cancer. First, USP13 promoted MCL1 protein stability in multiple lung and ovarian cancer cell models. Second, USP13 interacted with MCL1 and decreased MCL1 polyubiquitination in

a DUB activity-dependent manner. Third, USP13 and MCL1 were similarly amplified and overexpressed in lung and ovarian tumors. Finally, USP13 inhibition reduced MCL1 protein abundance and thereby increased tumor cell sensitivity to BH3 mimetic inhibitor ABT-263. Of note, re-expression of MCL1 upon USP13 depletion only partially rescued the xenograft growth in vivo, indicating a pleiotropic role of USP13 in regulating multiple oncoproteins. Taken together, USP13 may have context-dependent functions and the physiopathological significance of USP13 during oncogenesis requires mechanistic investigations on its relevant substrates.

Importantly, in addition to apply genetic approaches to modulate USP13 expression, we have tested a preclinical lead compound spautin-1 and evaluated the impact of pharmacologically inhibiting USP13 activity on MCL1 protein expression and tumor cell sensitivity to ABT-263. Spautin-1 is an inhibitor of the deubiquitinating activity of USP10 and USP13[40], and we observed no changes of USP13 protein levels associated with spautin-1 treatment. On the other hand, MCL1 progressively decreased upon spautin-1 exposure in a dose- and time-dependent manner,

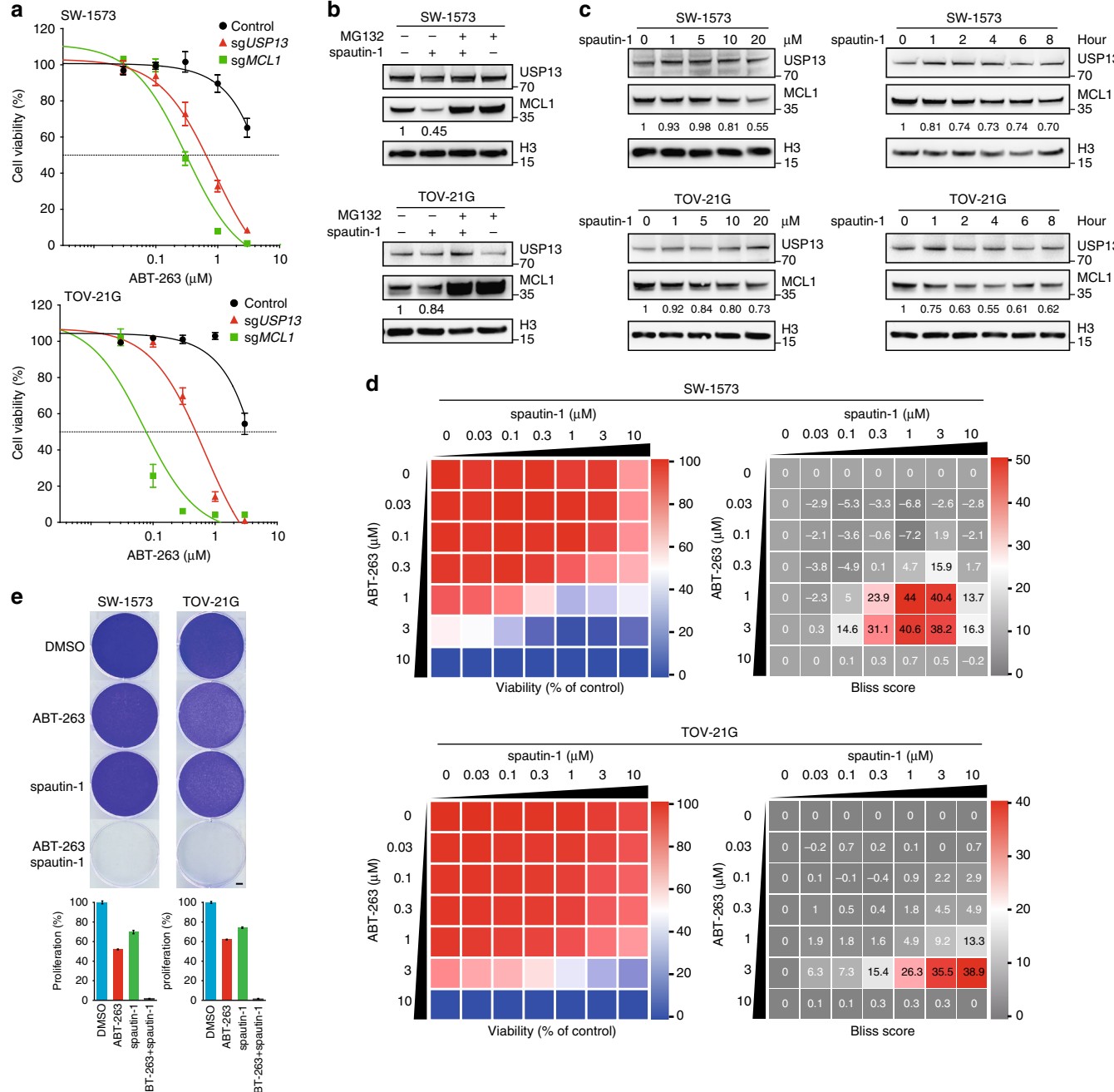

**Fig. 6** Targeting USP13 sensitizes tumor cells to BH3 mimetics. **a** *USP13* or *MCL1* was knocked out in SW-1573 and TOV-21G cells using CRISPR/Cas9, and cell viability in the presence of ABT-263 was analyzed by Cell Counting Kit-8. Error bars indicated standard deviation (each condition contained three biological replicates). **b** SW-1573 and TOV-21G cells were treated with spautin-1 (10 μM) and MG132 (10 μM) for 6 h. Cell lysates were analyzed by western blotting using antibodies against USP13, MCL1 or H3. **c** Cells were treated with spautin-1 at indicated concentrations or for different time frames, and cell lysates were analyzed by western blotting using antibodies against USP13, MCL1 or H3. **d** Examples of synergistic activities of spautin-1 and ABT-263 in SW-1573 and TOV-21G cells. Left: heatmaps of growth inhibition; right: heatmaps of Bliss synergy scores. **e** Cells were treated with DMSO, spautin-1 (1 μM) and/or ABT-263 (1 μM). The remaining cells were stained with crystal violet and quantified (scale bar=3.5 mm). Error bars indicated standard deviation (each condition contained three biological replicates)

presumably due to USP13 inhibition. As a result, tumor cells showed increased sensitivity toward ABT-263. These results confirmed that USP13 regulated MCL1 stability as a deubiquitinase, and further illustrated the feasibility of developing drug-like USP13 inhibitors for cancer management. Such therapeutics may be proven particularly valuable when used in synergy with BH3 mimetic compounds or selected chemotherapeutic agents, the efficacy of which is largely dependent on MCL1 levels. Although

recent development of MCL1 inhibitors such as S63845 suggests that the specific targeting of MCL1 in vivo is possible[31], the optimal clinical use is yet to be determined, considering that one inevitable concern regarding inhibitors of the BCL-2 family proteins is the potential unacceptable toxicity associated with complete functional shutdown[24, 47], especially when combined with other cytotoxic agents to treat solid tumors. On the other hand, USP13 displays increased activity in human cancers and

may represent a more titratable and episodic strategy for MCL1 inhibition to selectively kill tumor cells. In addition, it has been reported that mTOR inhibitors suppress cap-dependent translation of MCL1 mRNA[48, 49]. These different therapeutic approaches to target MCL1 use diverse mechanisms of action and warrant further investigations in clinic. Our study has provided a rational basis for designing clinical trials of USP13 inhibitors in the future to control solid tumors including lung and ovarian cancer.

In conclusion, we identified USP13 as a novel deubiquitinase to stabilize MCL1 and modulate tumor cell sensitivity to BH3 mimetic inhibitors. Hence, pharmaceutical intervention of USP13 activity is expected to antagonize the tumorigenic potential of MCL1 oncoprotein, and combined administration of USP13 inhibitors with clinically approved venetoclax therapy may represent a promising targeting strategy for the treatment of human cancer by inducing tumor cell death.

## Methods

**Cell culture and reagents**. The HEK293T and tumor cell lines were originally obtained from ATCC or JCRB in 2014, where mycoplasma contamination was tested and cell characterization was performed using polymorphic short tandem repeat (STR) profiling. Cells were cultured in RPIM 1640 (Life Technologies) supplemented with 10% fetal bovine serum (Millipore). ABT-263, spautin-1 and other inhibitors were purchased from Selleck Chemicals. All inhibitors were reconstituted in DMSO (Sigma-Aldrich) at a stock concentration of 10 mM. For cell viability assays, cells were seeded at 100,000 cells per well in growth media supplemented with 10% fetal bovine serum in 6-well plates, allowed to adhere overnight, and treated with a serial dilution of inhibitors for 5 days. Cells were fixed with formalin and stained with crystal violet.

**Plasmids, siRNA and sgRNA**. Plasmids were transfected into cells using Lipofectamine 2000 (Invitrogen). The USP13-C345A mutant was generated using a Q5® Site-Directed Mutagenesis Kit (New England Biolabs) according to the manufacturer's instructions. The primers used for generating the USP13-C345A mutant were: Forward-GAACCTGGGCAACAGCGCCTATCTCAGCTCTGTC; Reverse-GACAGAGCTGAGATAGGCGCTGTTGCCCAGGTTC.

All siRNAs were used at a final concentration of 25 nM and transfected into cells with Lipofectamine RNAiMAX reagent according to manufacturer's instructions (Invitrogen). For siRNA screen, a mixture of four siRNA sequences (Dharmacon) targeting each DUB was transfected into HEK293T cells. For FBXW7 knockdown, the siRNA sequences were: FBXW7-siRNA-1 GGAUCUCUUGAUACAUCAA; FBXW7-siRNA-2 GGAGUAUGGUCAUCACAAA.

USP13 and MCL1 were genetically knocked out by using the CRISPR/Cas9 system. The sgRNA sequences used for knockout were as follows: EGFP-sgRNA-F-CACCGGAAGTTCGAGGGCGACACCC; EGFP-sgRNA-R-AAACGGG TGTCGCCCTCGAACTTCC; USP13-sgRNA-1F-CACCGTTCCTGCCT CCGCTGCCGCC; USP13-sgRNA-1R-AAACGGCGGCAGCGGAGGCAGGAAC; USP13-sgRNA-2F-CACCGACGCGGATCGTGGGCATGTG; USP13-sgRNA-2R-AAACCACATGCCCACGATCCGCGTC; MCL1-sgRNA-1F-CACCGCCTCA CGCCAGACTCCCGGA; MCL1-sgRNA-1R-AAACTCCGGGAGTCT GGCGTGAGGC; MCL1-sgRNA-2F-CACCGTCTCCTCAAGCGGCGCCGCG; MCL1-sgRNA-2R-AAACCGCGGCGCCGCTTGAGGAGAC; HUWE1-sgRNA-1F-CACCGACCTGTTGGACCGCTTCGA; HUWE1-sgRNA-1R-AAACTCGAA GCGGTCCAACAGGTC; HUWE1-sgRNA-2F-CACCGGACCGCTTCGA TGGAATAC; HUWE1-sgRNA-2R-AAACGTATTCCATCGAAGCGGTCC.

**Virus production and infection**. HEK293T cells in a 10-cm dish were co-transfected with 5 μg of sgRNA constructs, 5 μg of plasmid Δ8.9, and 3 μg of plasmid VSVG. Cells were incubated at 37 °C and the medium was replaced after 12 h. Virus-containing medium was collected 48 h after transfection and supplemented with 8 μg mL⁻¹ polybrene to infect target cells in 6-well dishes. Infected cells were selected with 2–5 μg mL⁻¹ puromycin.

**Immunoprecipitation assays**. HEK293T cells were transfected with 3 × FLAG-USP13 for 48 h and were lysed with IP lysis buffer containing 50 mM Tris-HCl (pH 8.0), 150 mM NaCl, 10 mM KCl, 1.5 mM MgCl₂, 0.5% NP-40, 10% glycerol, 1 mM EDTA and protease inhibitor cocktail (Roche). The soluble lysate was pre-cleared with protein G beads and then 3 × FLAG-USP13 was immunoprecipitated with anti-FLAG M2 beads at 4 °C overnight. The beads were washed five times with IP lysis buffer and 3 × FLAG-USP13 was eluted with 500 μg mL⁻¹ 3 × FLAG peptide. Endogenous USP13 or MCL1 immunoprecipitation was performed in HEY, A549 and SW-1573 cells using 2 μg USP13 (Santa Cruz, sc-514416) or MCL1 antibodies (Santa Cruz, sc-819) and protein A/G agarose beads (Sigma) at 4 °C. The

immunocomplexes were then washed five times with IP lysis buffer. Both lysates and immunoprecipitates were analyzed by immunoblotting. For in vitro immunoprecipitation assay, 1 μg His-MCL1 (Prospec) and 2 μg His-USP13 (R&D systems) were incubated in 20 μL IP lysis buffer with or without the following components: 50 μM human ubiquitin, 100 nM UBE1, 2 μM UbE2D3, 2 μM CHIP, 2 μM Hsp70, 2 mM ATP (Ubiquitin-Proteasome Biotechnologies) at 37 °C for 2 h. After addition of 2 μg USP13 antibody (Santa Cruz, sc-514416), the samples were rotated at 4 °C overnight and incubated with protein A/G agarose beads (Sigma) for a further 4 h at 4 °C. The beads were washed five times with IP lysis buffer. The proteins were released from the beads by boiling in SDS-PAGE sample buffer and analyzed by immunoblotting.

**Western blot analysis**. Cells were lysed in RIPA buffer (Tris pH 7.4 50 mM, NaCl 150 mM, NP-40 1%, SDS 0.1%, EDTA 2 μM) containing proteinase inhibitors (Roche) and phosphatase inhibitors (Roche). The cell lysates (~20 μg protein) were quantified using a BCA Protein Assay Kit (Thermo Scientific) and subjected to SDS–PAGE followed by Western blot. Antibodies against the following proteins were used: USP13 (sc-514416, 1:1000), MCL1 (sc-819, 1:1000) (Santa Cruz); USP9X (#14898, 1:1000), PTEN (#9188, 1:1000), MULE (#5695, 1:1000), BCL-XL (#2764, 1:1000), BAX (#5023, 1:1000), BAK (#6947, 1:1000), Vimentin (#5741, 1:1000), GAPDH (#8884, 1:1000), H3 (#12648, 1:2000), β-tubulin (#5346, 1:1000), Actin (#5125, 1:1000) (Cell Signaling Technology); OTUB2 (ab74371, 1:1000) (Abcam). FLAG-tagged proteins were detected with a rabbit anti-FLAG antibody (Sigma, F3165, 1:1000). HA-tagged proteins were detected with a rabbit anti-HA antibody (Santa Cruz, sc-7392, 1:1000). For cellular fractionation, TOV-21G, OVCA433, HEY and NCI-H1581 cells were fractionated with the Qproteome Cell Compartment Kit (Qiagen) and equivalent amounts of fractionated cell lysates were analyzed by immunoblotting. Uncropped images of the immunoblots were shown in Supplementary Figs. 7 and 8.

**Immunofluorescence microscopy**. HEY cells were fixed for 15–20 min in 4% paraformaldehyde and permeabilized with 0.1% Triton X-100 (in PBS) for 10 min. After three PBS washes, cells were blocked with 2% BSA for 30 min at room temperature. Cells were then incubated with anti-USP13 antibody (Santa Cruz, sc-514416, 1:500) and anti-MCL1 antibody (Santa Cruz, sc-819, 1:500) diluted in 2% BSA at 4 °C overnight. After three PBS washes, the cells were incubated with 1 μg mL⁻¹ Alexa Fluor 488/594-conjugated secondary antibodies (Invitrogen) shaking at room temperature in the dark for 1 hour. Cells were washed three times with PBS in the dark, stained with DAPI (Invitrogen) and mounted in Prolong® Gold Antifade Reagent (Invitrogen). For mitochondrial morphology analysis, SW-1573 and TOV-21G cells were grown on coverslips inside a Petri dish and incubated with pre-warmed staining solution containing 50 nM MitoTracker Red CMXRos (Invitrogen) for 45 min at 37 °C. The immunofluorescent staining was observed using a confocal microscope (Leica).

**Immunohistochemistry**. The lung cancer tissue microarray, which contained 30 cases of lung adenocarcinomas and their adjacent normal lung tissues, was purchased from Shanghai Outdo Biotech Company. The ovarian cancer specimens were collected during surgery in Ren Ji Hospital, Shanghai Jiao Tong University. The tissue slides were deparaffinized, treated with 3% H₂O₂ for 10 min, autoclaved in 10 mM citric sodium (pH 6.0) for 30 min to unmask antigens, rinsed in PBS, and then incubated with the primary antibodies against MCL1 (Santa Cruz, sc-819, 1:100) or USP13 (Santa Cruz, sc-514416, 1:250) at 4 °C overnight, followed by incubation with biotinylated secondary antibody for 1 hour at room temperature. Signal amplification and detection was performed using the DAB system according to the manufacturer's instructions (Dako). Statistical significances of the correlation between USP13 and MCL1 expressions or the association of USP13 or MCL1 protein levels with tissue types (normal versus cancer) were determined using a χ² (chi-squared) test. R represented the correlation coefficients.

**Deubiquitylation assays**. For in vivo MCL1 ubiquitylation assay, USP13, MCL1 and HA-ubiquitin plasmids were transfected into HEK293T cells using Lipofectamine 2000. After 48 h, the cells were treated with 20 μM proteasome inhibitor MG132 (Selleck) for 8 h. The cells were then washed with PBS and lysed in HEPES buffer (20 mM HEPES, pH 7.2, 50 mM NaCl, 1 mM NaF, 0.5% Triton X-100) supplemented with 0.1% SDS, 20 μM MG132 and protease inhibitor cocktail (Roche). The lysates were centrifuged and incubated with 2 μg anti-MCL1 antibody (Santa Cruz, sc-819) overnight and protein A/G agarose beads (Sigma) for a further 8 h at 4 °C. The beads were washed three times with HEPES buffer. The proteins were released from the beads by boiling in SDS–PAGE sample buffer and analyzed by immunoblotting with an anti-HA antibody (Santa Cruz, sc-7392, 1:1000). For in vitro MCL1 ubiquitylation assay, the reaction was carried out at 37 °C for 2 h in 20 μL reaction buffer (20 mM Tris-HCl, pH 7.2, 5 mM MgCl₂, 50 mM NaCl, 1 mM 2-mercaptoethanol, 10% glycerol) containing the following components: 50 μM human ubiquitin, 100 nM UBE1, 2 μM UbE2D3, 2 μM CHIP, 2 μM Hsp70, 2 mM ATP (Ubiquitin-Proteasome Biotechnologies), 5 μg His-MCL1 (Prospec) and 5 μg His-USP13 (R&D systems). After addition of 2 μg MCL1 antibody (Santa Cruz, sc-819), the samples were rotated at 4 °C overnight and incubated with protein A/G agarose beads (Sigma) for a further 4 h at 4 °C. The beads were washed three times

with HEPES buffer. The proteins were released from the beads by boiling in SDS–PAGE sample buffer and analyzed by immunoblotting.

**Quantitative PCR assays.** Total RNA from tumor cells was extracted with Trizol (Invitrogen) and subjected to reverse transcription using the High-Capacity cDNA Reverse Transcription Kit (Invitrogen). The resultant cDNA was used to perform the quantitative PCR on the Applied Biosystems ViiA7 machine. Relative expression levels of each gene were normalized to *GAPDH* as the endogenous control for all experiments. At least three biological replicates were included for each condition.

**Cell migration assays.** Transwell chambers with 8 µm pore membranes (Corning) placed in 24-well culture plates were incubated with serum-free RPMI 1640 medium at 37 °C for 1 h. A 200 µL suspension of $0.5–1 \times 10^5$ cells was seeded in the upper compartment of the transwell chambers with 600–800 µL RPMI 1640 medium supplemented with 10% fetal bovine serum in the bottom wells, and incubated at 37 °C for 12–24 h. The migrating cells attached to the lower membranes of transwell chambers were stained with crystal violet and counted.

**Cell viability assays and combination matrices.** Cell proliferation assay was performed by using Cell Counting Kit-8 (Dojindo Laboratories). Cells were seeded in triplicate in 96-well plates and subjected to the indicated treatments for 96 h before measuring the absorbance at 450 nm according to the manufacturer's instructions. Cell apoptosis was detected by measuring activation of caspase-3/7 using the Caspase-Glo 3/7 Assay kit (Promega). For combination matrices, cells were seeded in 96-well plates at 3000–5000 cells per well. After 24 h, cells were treated with ABT-263 (dose range of 0–10 µM) and spautin-1 (dose range of 0–10 µM) in a $7 \times 7$ matrix. Cells were cultured with inhibitors for 96 h and cell viability was determined using Cell Counting Kit-8. Each treatment was done in triplicate wells. The Bliss expectation was calculated by the equation $(A+B)–A \times B$. A and B were the fractional growth inhibitions induced by agents A and B at a given dose.

**Xenograft models.** Tumor cells ($1 \times 10^6$) were mixed with Matrigel (BD Biosciences) and subcutaneously implanted in the dorsal flank of BALB/c Nude mice. Tumor volumes (8 animals per group) were measured with a caliper and calculated as length×width$^2$×0.52. The institutional animal care and use committee of Ren Ji Hospital approved all animal protocols.

**Statistical analysis.** In all experiments, comparisons between two groups were based on two-sided Student's *t*-test and one-way analysis of variance (ANOVA) was used to test for differences among more groups. *P*-values of < 0.05 were considered statistically significant.

**Data availability.** All data are included within the article or available from the authors upon request.

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

## Acknowledgements

This work was supported by the National Natural Science Foundation of China (81472537 and 81672714 to G.Z.; 81502597 to Y.J.; 81472426 to W.D.), the Grants from the State Key Laboratory of Oncogenes and Related Genes (No. 91-15-12 to G.Z.), the grants from Shanghai Jiao Tong University School of Medicine (DLY201505 to W.D.; YG2016MS51 to X.Y.), Shanghai Municipal Education Commission-Gaofeng Clinical Medicine Grant Support (20161313 to G.Z.), the Shanghai Institutions of Higher Learning (Eastern Scholar to G.Z.), Shanghai Rising-Star Program (16QA1403600 to G.Z.), Shanghai Municipal Commission of Health and Family Planning (2013ZYJB0202 and 15GWZK0701 to W.D.; 20174Y0189 to Y.J.), the grant from Shanghai Key Laboratory of Gynecologic Oncology (FKZL-2017-01 to Y.J.), and the grant from Science and Technology Commission of Shanghai Municipality (16140904401 to X.Y.).

## Author contributions

S.Z., M.Z., and Y.J. designed and conceived the experiments. X.Y., P.M., and Z.Z. analyzed the data. X.W. and W.D. provided clinical samples. S.Z. and G.Z. drafted the manuscript and all authors contributed to writing the manuscript. G.Z. supervised the study.

## Additional information

**Competing interests:** The authors declare no competing financial interests.

