## [Peer Review File · Nature Communications]

Reviewer #1 (Remarks to the Author):

The authors describe an 84-gene screen using siRNAs targeting deubiquitinases (DUBs) to identify DUBs where knockdown decreases the stability of the MCL-1 protein. While earlier studies identified USP9X as a DUB that stabilizes protein levels of MCL-1, this study finds that USP13 stabilizes protein level of MCL-1 in lung and ovarian cancer cells. Copy number analysis and immunohistochemistry were performed in human samples to indicate that USP13 may be a relevant target in ovarian and lung cancer. The authors show that knockout of USP13 sensitizes cancer cells to navitoclax – an inhibitor of BCL-2 and BCL-XL., and that knockout of USP13 delays or prevents tumor growth in cell line xenografts. Although the study suggests that USP13 would be an interesting therapeutic target in lung and ovarian cancer, a more thorough pre-clinical evaluation of the utility in targeting USP13 would increase enthusiasm and the potential influence of the paper.

Overall, I think that they likely have identified USP13 as another DUB that affects MCL-1 stability in some contexts. Its therapeutic relevance is not so clear, as Bh3 mimetic inhibition of MCL-1 seems more direct and narrow, and even CDK9 inhibition might be more specific to regulating MCL-1. Some major and minor issues are listed below:

Major Issues

- In addition to its role as an anti-apoptotic protein, MCL-1 appears to have a role in normal physiology for mitochondrial fusion, respiration, and other mitochondrial phenotypes (Perciavalle et al., Nature Cell Biology 2012). Broad destabilization of MCL-1 may therefore have negative results on normal cells, and may not be a great therapeutic strategy. It would be useful if the authors could determine whether USP13 knockdown or knockouts alters mitochondrial fusion or respiration.
- Another paper in Nature Communications from 2016 (cited by the authors - reference 43) indicates that knockdown/knockout of USP13 delays or prevents tumor growth by modifying tumor metabolism. It's unclear whether the modification to tumor metabolism or destabilization of MCL-1 is responsible for the modification to tumor progression. It would be useful if the authors could show that MCL-1 destabilization contributes to the delay or tumor growth by performing experiments where there is simultaneous knockout of USP13 and expression of a MCL-1 mutant that either lacked the residues required for ubiquitination (either by deletion or mutation). Loss of USP13 might do a lot of things. Since MANY cellular perturbations might affect efficiency of transcription and translation, and MCL-1 is a short half-life protein and transcript, the direct regulation of MCL-1 as the MOA of decreased tumor fitness is not clear.
- The authors comment on the tissue specific expression of USP9x limiting its significance in regulating MCL-1. Can the same be said for USP13? What is its tissue specificity beyond ovarian and lung cancer?
-

Minor Issue:

- Perhaps the authors could provide more discussion on the differences in targeting USP13 to destabilize MCL-1 rather than using on-target inhibitors like S63845 which appears to have some effectiveness in cancers in mice.

Reviewer #2 (Remarks to the Author):

In the present study, Zhang et al. could identify USP13 as a specific DUB for regulating MCL1 stability in lung and ovarian cancer cells. Both USP13 and MCL1 were found to be genomically amplified and upregulated in human lung and ovarian tumor tissues. At the molecular level, USP13 was shown to bind to MCL1 and stabilize MCL1 protein levels via a deubiquitination mechanism. Downregulation or pharmacological inhibition of USP13 inhibited tumor growth in xenograft animal model and reduced MCL1 protein levels, respectively. This is an interesting finding suggesting a novel function for USP13. However, I have several concerns about the experimental design, interpretation of data, and the significance of the results.

The major concern is the hypothesis that

a) MCL1 and USP13 together are important factor for tumor development/progression. This might be explain by constitutive ubiquitination of MCL1 by FBW7/MULE, and in this scenario, the high expression of USP13 prevents ubiquitination ad degradation of MCL1. If this is the correct hypothesis, the authors need to investigate this issue thoroughly.

b) Another weakness of the manuscript is that the control experiments investigating the mechanism of USP13 deubiquitination of MCL1 stability has no effect in the cell survival/apoptosis or proliferation.

c) Hypoxia and oxidative stress are independent mechanism that causes upregulation/downregulation of diverse genes and signaling pathway probably not entirely connected to USP13 binding to MCL1.

d) Finally, it is not clear whether USP13-MCL1 association is dependent/independent of MCL1 ubiquitination.

Major comments:

1. In the manuscript it has suggested that MCL1, which is an anti-apoptotic protein together with USP13 are important factors for cancer development and progression. In this aspect the levels of MULE and FBW7 need to be analyzed and explain whether these E3 ligases causing constitutive ubiquitination of MCL1 in cancer cells, which is further reversed by USP13.

2. The authors found that USP13 and MCL1 are genomically amplified and upregulated only at the protein but not at the RNA. What is the explanation that genome amplification increases the levels of these proteins without any changes at the RNA levels?

3. In the introduction it has been stated that inhibition of DUBs are considered as attractive therapeutic targets for cancer, however up to date the majority of DUBs are actually acting as tumor suppressor genes rather than oncogene.

4. The authors argue that the aim of the study is to identify the ubiq/deubiq. mechanism of MCL1 in cancer. In order to identify the specific DUB for USP13, a deubiquitinase knockdown screening is performed. If the aim of the study was to understand the function of MCL1 in cancer, what is the argument of using HEK293 cells (embryonic kidney cells) for this type of screening? Has the authors check whether MCL1 is constitutive ubiq. in HEK293 cells similar to lung and ovarian cancer and that a reduction of USP13 elevates the levels of MCL1 in these cells? Does HEK293 cells express USP13 and USP9X at all?

5. In manuscript, 22 lung and 33 ovarian cancer cell lines were used but the origin of these cells is not explained in the methods.

6. The hypothesis that high USP13 expression leads to elevated levels of MCL1 is not supported by the western blot shown in Fig. 2C especially for lung cancer cell lines. Many cell lines have high levels of USP13 and low levels of MCL1 or vice versa.

7. The result presented in figure 1E is not consistent. It is a significant variation among the siRNAs tested for USP9X and OTUB2 concerning the expression levels of MCL1.

8. Fig 3A, does treatment of siUSP13 cells with MG132 rescue the levels of MCL1?

9. Figure 4; the authors suggests an interaction between MCL1 and USP13. The question is whether this interaction is direct or via another binding protein. Direct interaction/binding assay need to confirm this finding.

10. The results discussed in figure 4E is not clear and the data is not convincing at all. In addition, an in vitro deubiq. assay need to be performed to ascertain that MCL1 is a specific substrate for USP13.

11. Figure 5A; it is concluded that USP13 depletion reduces MCL1 protein levels. The results and blots presented in this figure is not convincing.

12. As it was discussed earlier, hypoxia-mediated death induction is a robust mechanism leading to expression of different genes that could causes the results shown in Figure 5B independent of the MCL1-USP13 axes.

13. The part of the study dealing with the inhibitor ABT-263 is very confusing and it is very difficult to follow the arguments such as “Because drug development specifically against MCL1 has been considerably slower, targeting USP13 to inhibit MCL1 indirectly is likely to represent an alternative approach to potentiate the therapeutic efficacy of ABT-263 in cancer cells”.

14. It is not clear what figure 6E is showing.

15. Fig. 1A is not necessary. This is a very simple and not complicated protocol.

Reviewers' Comments to Author

Reviewer #1:

Comments to the Author

The authors describe an 84-gene screen using siRNAs targeting deubiquitinases (DUBs) to identify DUBs where knockdown decreases the stability of the MCL-1 protein. While earlier studies identified USP9X as a DUB that stabilizes protein levels of MCL-1, this study finds that USP13 stabilizes protein level of MCL-1 in lung and ovarian cancer cells. Copy number analysis and immunohistochemistry were performed in human samples to indicate that USP13 may be a relevant target in ovarian and lung cancer. The authors show that knockout of USP13 sensitizes cancer cells to navitoclax – an inhibitor of BCL-2 and BCL-XL., and that knockout of USP13 delays or prevents tumor growth in cell line xenografts. Although the study suggests that USP13 would be an interesting therapeutic target in lung and ovarian cancer, a more thorough pre-clinical evaluation of the utility in targeting USP13 would increase enthusiasm and the potential influence of the paper.

Overall, I think that they likely have identified USP13 as another DUB that affects MCL-1 stability in some contexts. Its therapeutic relevance is not so clear, as Bh3 mimetic inhibition of MCL-1 seems more direct and narrow, and even CDK9 inhibition might be more specific to regulating MCL-1. Some major and minor issues are listed below:

Major Issues

- In addition to its role as an anti-apoptotic protein, MCL-1 appears to have a role in normal physiology for mitochondrial fusion, respiration, and other mitochondrial phenotypes (Perciavalle et al., Nature Cell Biology 2012). Broad destabilization of MCL-1 may therefore have negative results on normal cells, and may not be a great therapeutic strategy. It would be useful if the authors could determine whether USP13 knockdown or knockouts alters mitochondrial fusion or respiration.

We thank the reviewer for the excellent suggestions. The mitochondrial fusion has been determined by MitoTracker fluorescent dye to stain mitochondria in live tumor cells. We found that USP13 knockout or knockdown did not lead to altered mitochondrial fusion. These data have been provided in the revised manuscript (Supplementary Fig. 5c).

- Another paper in Nature Communications from 2016 (cited by the authors - reference 43) indicates that knockdown/knockout of USP13 delays or prevents tumor growth by modifying tumor metabolism. It's unclear whether the modification to tumor metabolism or destabilization of MCL-1 is responsible for the modification to tumor progression. It would be useful if the authors could show that MCL-1 destabilization contributes to the delay or tumor growth by performing experiments where there is simultaneous knockout of USP13 and expression of a MCL-1 mutant that either lacked the residues required for

ubiquitination (either by deletion or mutation). Loss of USP13 might do a lot of things. Since MANY cellular perturbations might affect efficiency of transcription and translation, and MCL-1 is a short half-life protein and transcript, the direct regulation of MCL-1 as the MOA of decreased tumor fitness is not clear.

The reviewer raised an excellent point. We performed the rescue experiment by overexpressing MCL1 in USP13-depleted tumor cells (SW-1573). Reassuringly, MCL1 expression resulted in a partial recovery of xenograft growth in nude mice (Fig. 5f), indicating that MCL1 was a functionally crucial USP13 substrate. Notably, the rescue was not complete as evidenced by smaller tumor volume, which suggested that other mechanisms (e.g. metabolic regulators) might be also responsible for the deficient tumor development as a result of USP13 deletion. We have included the new data and discussion in the revised manuscript.

• The authors comment on the tissue specific expression of USP9x limiting its significance in regulating MCL-1. Can the same be said for USP13? What is its tissue specificity beyond ovarian and lung cancer?

We analyzed the expression landscape of USP13 mRNA or protein using The Human Protein Atlas. USP13 was ubiquitously expressed in different human tissues as well as various cancer types, consistent with increased copy numbers.

mRNA EXPRESSION

PROTEIN EXPRESSION

Minor Issue:

- Perhaps the authors could provide more discussion on the differences in targeting USP13 to destabilize MCL-1 rather than using on-target inhibitors like S63845 which appears to have some effectiveness in cancers in mice.

We appreciate the reviewer's great point. Recent development of MCL1 inhibitors such as S63845 suggests that the specific targeting of MCL1 *in vivo* is possible, but their optimal clinical use is yet to be determined. One inevitable concern regarding inhibitors of the BCL-2 family proteins is the potential unacceptable toxicity associated with complete functional shutdown, especially when S63845 has to be combined with other agents to treat solid tumors. On the other hand, USP13 displays increased activity in human cancer and may represent a more titratable and episodic strategy for MCL1 inhibition to selectively kill tumor cells. Additionally, it has been reported that mTOR inhibitors suppress cap-dependent translation of MCL1 mRNA. These different approaches to target MCL1 use diverse mechanisms of action and warrant further investigations in clinic. We have provided more discussion in the revised manuscript.

Reviewer #2:

Comments to the Author

In the present study, Zhang et al. could identify USP13 as a specific DUB for regulating MCL1 stability in lung and ovarian cancer cells. Both USP13 and MCL1 were found to be genomically amplified and upregulated in human lung and ovarian tumor tissues. At the molecular level, USP13 was shown to bind to MCL1 and stabilize MCL1 protein levels via a deubiquitination mechanism. Downregulation or pharmacological inhibition of USP13 inhibited tumor growth in xenograft animal model and reduced MCL1 protein levels, respectively. This is an interesting finding suggesting a novel function for USP13. However, I have several concerns about the experimental design, interpretation of data, and the significance of the results.

The major concern is the hypothesis that

a) MCL1 and USP13 together are important factor for tumor development/progression. This might be explain by constitutive ubiquitination of MCL1 by FBW7/MULE, and in this scenario, the high expression of USP13 prevents ubiquitination ad degradation of MCL1. If this is the correct hypothesis, the authors need to investigate this issue thoroughly.

We thank the reviewer for the insightful comments. The experiment to investigate the impact of ubiquitin ligases on MCL1 in the context of USP13 regulation has been performed. Both FBW7 and MULE played an important part in modulating MCL1 expression, and USP13 was required to reverse the ubiquitination and degradation of MCL1. These results were described below in detail.

b) Another weakness of the manuscript is that the control experiments investigating the mechanism of USP13 deubiquitination of MCL1 stability has no effect in the cell survival/apoptosis or proliferation.

We have analyzed different tumor cell models in various experimental settings, and discovered that neither cell proliferation nor migration was significantly changed by USP13 depletion under normal conditions. However, in the presence of hypoxia, oxidants or BH3 mimetic inhibitors, USP13 loss caused a substantial decrease of cancer cell viability. The effect was most likely dependent on the MCL1 downregulation, as MCL1 overexpression apparently rescued tumor expansion. Therefore, the USP13-MCL1 axis could have a specific function to overcome the hostile growth environments, which presumably induce cell apoptosis.

c) Hypoxia and oxidative stress are independent mechanism that causes upregulation/downregulation of diverse genes and signaling pathway probably not entirely connected to USP13 binding to MCL1.

We totally agree with the reviewer and the manuscript did not intend to pinpoint USP13-MCL1 as a unique signaling pathway in response to hypoxic and oxidative stresses. Rather, we propose that the stabilization of MCL1 by USP13 may play an important role in the contexts of various pro-apoptotic insults, such as those elicited by hypoxia, oxidants or BH3 mimetic inhibitors. We have included this point in the discussion.

d) Finally, it is not clear whether USP13-MCL1 association is dependent/independent of MCL1 ubiquitination.

These critical data have been added to the revised manuscript, and the results were described below in detail.

Major comments:

1. In the manuscript it has suggested that MCL1, which is an anti-apoptotic protein together with USP13 are important factors for cancer development and progression. In

this aspect the levels of MULE and FBW7 need to be analyzed and explain whether these E3 ligases causing constitutive ubiquitination of MCL1 in cancer cells, which is further reversed by USP13.

We appreciate the reviewer's insightful comments. Following these suggestions, MULE and FBW7 was knocked down individually or in combination in SW-1573 (lung cancer) and HEY (ovarian cancer) cells. We found that both FBW7 and MULE played an important role in regulating MCL1 protein levels, and USP13 was able to reverse the ubiquitination and degradation of MCL1. However, when both MULE and FBW7 were absent, the protective effects of USP13 were diminished, indicating that USP13 functioned against the E3 ubiquitin ligases. These results were provided in the revised manuscript (Supplementary Fig. 3d).

2. The authors found that USP13 and MCL1 are genomically amplified and upregulated only at the protein but not at the RNA. What is the explanation that genome amplification increases the levels of these proteins without any changes at the RNA levels?

We apologize for the confusion. Both USP13 and MCL1 were genomically amplified in lung and ovarian cancers, and their mRNA expression did correlate with the copy number status. However, we observed that within each cancer, the USP13 expression significantly correlated the MCL1 expression only at the protein, but not mRNA, levels.

3. In the introduction it has been stated that inhibition of DUBs are considered as attractive therapeutic targets for cancer, however up to date the majority of DUBs are actually acting as tumor suppressor genes rather than oncogene.

We believe that DUBs may serve as oncogenes or tumor suppressors, based on the functional substrates that they regulate. There are many examples in the literature showing that some DUBs have tumorigenic role and thus may be potential therapeutic targets, such as USP9X (Schwickart et al., Nature. 2010), OTUD7B (Wang et al., Nature. 2017), YOD1 (Kim et al., Proc Natl Acad Sci U S A. 2017), USP36 (Sun et al., Proc Natl Acad Sci U S A. 2015), and USP28 (Popov et al., Nat Cell Biol. 2007). We have included more references to support our point.

4. The authors argue that the aim of the study is to identify the ubiq/deubiq. mechanism of MCL1 in cancer. In order to identify the specific DUB for USP13, a deubiquitinase knockdown screening is performed. If the aim of the study was to understand the function of MCL1 in cancer, what is the argument of using HEK293 cells (embryonic kidney cells) for this type of screening? Has the authors check whether MCL1 is constitutive ubiq. in HEK293 cells similar to lung and ovarian cancer and that a reduction of USP13 elevates the levels of MCL1 in these cells? Does HEK293 cells express USP13 and USP9X at all?

We used HEK293T cells for the siRNA screen, taking advantage of the high transfection efficiency. HEK293T has been frequently used as a model system for knockdown screening or protein interaction studies, in order to generate hypothesis to be further tested

in physiologically relevant systems. As shown in the revised Fig. 1d, HEK293T cells expressed detectable levels of USP13 and USP9X, and USP13 siRNAs decreased the endogenous protein levels of MCL1. The deubiquitination mechanism of MCL1 by USP13 held true in lung and ovarian cancer cells, as determined by our stringent validation experiments in Fig. 3.

5. In manuscript, 22 lung and 33 ovarian cancer cell lines were used but the origin of these cells is not explained in the methods.

We have provided the source of these cell lines in the revised Methods.

6. The hypothesis that high USP13 expression leads to elevated levels of MCL1 is not supported by the western blot shown in Fig. 2C especially for lung cancer cell lines. Many cell lines have high levels of USP13 and low levels of MCL1 or vice versa.

As we stated in the manuscript, the major conclusion of Fig. 2c was that most lung and ovarian cancer cell lines ubiquitously expressed high levels of USP13 and MCL1. We agree with the reviewer that the two proteins did not correlate well in tumor cell lines; however, they tended to positively correlate in human lung and ovarian cancer tissues. Two reasons may explain these seemingly paradoxical results. First, cell lines could display altered protein expression patterns during in vitro selection, such that the levels of USP13 and MCL1 were generally high. Second, other mechanisms might exist to simultaneously regulate MCL1 expression in cancer cells. Nevertheless, the correlation analysis in clinical specimens was more likely biologically relevant to understand the interaction between USP13 and MCL1.

7. The result presented in figure 1E is not consistent. It is a significant variation among the siRNAs tested for USP9X and OTUB2 concerning the expression levels of MCL1.

Yes, we indeed observed that modulating USP9X or OTUB2 with different siRNAs had no consistent effect on MCL1 expression. In contrast, all the four oligos against USP13 consistently decreased the endogenous protein levels of MCL1. Therefore, we concluded that USP13, instead of USP9X or OTUB2, might be a novel candidate deubiquitinase for MCL1 in these models.

8. Fig 3A, does treatment of siUSP13 cells with MG132 rescue the levels of MCL1?

We performed the experiment as suggested, and found that MG132 treatment did indeed largely rescue the protein levels of MCL1 (Supplementary Fig. 3b).

9. Figure 4; the authors suggests an interaction between MCL1 and USP13. The question is whether this interaction is direct or via another binding protein. Direct interaction/binding assay need to confirm this finding.

This is an excellent point. We have performed the in vitro immunoprecipitation

experiment to determine whether the interaction between MCL1 and USP13 is direct or indirect. As shown in Fig. 4c, USP13 could directly bind with MCL1, and the interaction was independent on MCL1 ubiquitination status.

10. The results discussed in figure 4E is not clear and the data is not convincing at all. In addition, an in vitro deubiq. assay need to be performed to ascertain that MCL1 is a specific substrate for USP13.

We appreciate the reviewer's excellent advice. The in vitro deubiquitination assay has been performed to support that MCL1 is a direct substrate of USP13 (Fig. 4g).

11. Figure 5A; it is concluded that USP13 depletion reduces MCL1 protein levels. The results and blots presented in this figure is not convincing.

We agree with the reviewer that the original blots are not convincing. A new figure has been provided with biological replicates and image quantification, which clearly shows that USP13 depletion by CRISPR/Cas9 reduces MCL1 protein levels.

12. As it was discussed earlier, hypoxia-mediated death induction is a robust mechanism leading to expression of different genes that could causes the results shown in Figure 5B independent of the MCL1-USP13 axes.

As we stated above, the manuscript did not intend to pinpoint USP13-MCL1 as the main signaling pathway that responds to hypoxic and oxidative stresses. Rather, we propose that the stabilization of MCL1 by USP13 may play an important role specifically in the contexts of various pro-apoptotic insults, such as those elicited by hypoxia, oxidants or BH3 mimetic inhibitors. We have further measured caspase-3/7 activities to quantify cell apoptosis upon USP13 loss (Fig. 5c). These new data support that the USP13-MCL1 axis has an anti-apoptotic function to overcome the hostile growth environments.

13. The part of the study dealing with the inhibitor ABT-263 is very confusing and it is very difficult to follow the arguments such as "Because drug development specifically against MCL1 has been considerably slower, targeting USP13 to inhibit MCL1 indirectly is likely to represent an alternative approach to potentiate the therapeutic efficacy of ABT-263 in cancer cells".

We apologize for the confusion due to insufficient explanation. MCL1 activity has been considered as one major mechanism of drug resistance to current BH3 mimetic inhibitors including ABT-263 and ABT-199, which target other BCL-2 members but spare the MCL1 molecule. Hence, MCL1 inhibition may potentiate the therapeutic efficacy of BH3 mimetic inhibitors against tumor cells, as advised by previous reports (Teh et al., Leukemia. 2017; Alford et al., Cancer Res. 2015; van Delft et al., Cancer Cell. 2006; Konopleva et al., Cancer Cell. 2006). In our study, we showed that genetic or pharmacological inactivation of USP13 could reduce MCL1 protein abundance and thus improve cell response to the BH3 mimetic inhibitor ABT-263. As the development of

small molecule inhibitors for MCL1 has been challenging, we think that targeting USP13 to indirectly inhibit MCL1 may be an alternative approach to increase ABT-263 sensitivity.

14. It is not clear what figure 6E is showing.

Fig. 6e was shown to validate the synergistic effect between ABT-263 and spautin-1 against tumor cells by using crystal violet staining, a way to directly visualize cell viability in the presence of drug treatments. We have also included the data quantification in the revised manuscript to make the point clearer.

15. Fig. 1A is not necessary. This is a very simple and not complicated protocol.

We have deleted Fig. 1a based on the reviewer's advice.

Reviewer #1 (Remarks to the Author):

The authors have made a satisfactory response to the comments. I think this will make an interesting addition to our knowledge of MCL-1 biology.

Reviewer #2 (Remarks to the Author):

The authors addressed the reviewer's comments.

Minor comments to the rebuttal letter:

1. We appreciate the reviewer's insightful comments. Following these suggestions, MULE and FBW7 was knocked down individually or in combination in SW-1573 (lung cancer) and HEY (ovarian cancer) cells. We found that both FBW7 and MULE played an important role in regulating MCL1 protein levels, and USP13 was able to reverse the ubiquitination and degradation of MCL1. However, when both MULE and FBW7 were absent, the protective effects of USP13 were diminished, indicating that USP13 functioned against the E3 ubiquitin ligases. These results were provided in the revised manuscript (Supplementary Fig. 3d).

Q: Downregulation exp. has been working well but I have difficulty to see degradation or any major changes in the levels of MCL1 upon downregulation of FBW7 and MULE alone or in combination with USP13.

2. We believe that DUBs may serve as oncogenes or tumor suppressors, based on the functional substrates that they regulate. There are many examples in the literature showing that some DUBs have tumorigenic role and thus may be potential therapeutic targets, such as USP9X (Schwickart et al., Nature. 2010), OTUD7B (Wang et al., Nature. 2017), YOD1 (Kim et al., Proc Natl Acad Sci U S A. 2017), USP36 (Sun et al., Proc Natl Acad Sci U S A. 2015), and USP28 (Popov et al., Nat Cell Biol. 2007). We have included more references to support our point.

Q: I agree with the comments provided by the authors, however, there are many DUBs that function as tumor suppressor gene, therefore I would suggest to revise the text to "a few DUBs..." and remove "may" from this sentence.

3. We appreciate the reviewer's excellent advice. The in vitro deubiquitination assay has been performed to support that MCL1 is a direct substrate of USP13 (Fig. 4g).

Q: It is very difficult to see any deubiq. of MCL1 in this assay. In the ubiq. blot one can see ubiq. of MCL1 from 50 kDa (the blot should show from 35 kDa. since the size of MCL1 is 35 kDa) up to 100kDa.

** See Nature Research's author and referees' website at www.nature.com/authors for information about policies, services and author benefits

Reviewers' Comments to Author

Reviewer #1:

Comments to the Author

The authors have made a satisfactory response to the comments. I think this will make an interesting addition to our knowledge of MCL-1 biology.

Reviewer #2:

Comments to the Author

The authors addressed the reviewer's comments.

Minor comments to the rebuttal letter:

1. We appreciate the reviewer's insightful comments. Following these suggestions, MULE and FBW7 was knocked down individually or in combination in SW-1573 (lung cancer) and HEY (ovarian cancer) cells. We found that both FBW7 and MULE played an important role in regulating MCL1 protein levels, and USP13 was able to reverse the ubiquitination and degradation of MCL1. However, when both MULE and FBW7 were absent, the protective effects of USP13 were diminished, indicating that USP13 functioned against the E3 ubiquitin ligases. These results were provided in the revised manuscript (Supplementary Fig. 3d).

Q: Downregulation exp. has been working well but I have difficulty to see degradation or any major changes in the levels of MCL1 upon downregulation of FBW7 and MULE alone or in combination with USP13.

Indeed, upon MULE or FBXW7 knockdown, the effects of USP13 on MCL1 were less visible, as compared to those in parental cancer cells (Fig. 3a). We speculated that the overall levels of MCL1 ubiquitination were decreased in the absence of MULE or FBXW7. To make the point clearer, we quantified the immunoblotting, which showed modest downregulation of MCL1 following USP13 knockdown (revised Supplementary Fig. 3d). Such a reduction was not observed when both MULE and FBXW7 were absent.

2. We believe that DUBs may serve as oncogenes or tumor suppressors, based on the functional substrates that they regulate. There are many examples in the literature showing that some DUBs have tumorigenic role and thus may be potential therapeutic targets, such as USP9X (Schwickart et al., Nature. 2010), OTUD7B (Wang et al., Nature. 2017), YOD1 (Kim et al., Proc Natl Acad Sci U S A. 2017), USP36 (Sun et al., Proc Natl Acad Sci U S A. 2015), and USP28 (Popov et al., Nat Cell Biol. 2007). We have included more references to support our point.

Q: I agree with the comments provided by the authors, however, there are many DUBs that function as tumor suppressor gene, therefore I would suggest to revise the text to "a

few DUBs...” and remove “may” from this sentence.

The reviewer’s point is well taken, and we have revised the text as suggested.

3. We appreciate the reviewer’s excellent advice. The in vitro deubiquitination assay has been performed to support that MCL1 is a direct substrate of USP13 (Fig. 4g).

Q: It is very difficult to see any deubiq. of MCL1 in this assay. In the ubiq. blot one can see ubiq. of MCL1 from 50 kDa (the blot should show from 35 kDa. since the size of MCL1 is 35 kDa) up to 100kDa.

We agree with the reviewer that our result for in vitro deubiquitination assay was not as obvious as that for in vivo deubiquitination assay, which was sometimes seen in literature as well (e.g., Inuzuka et al., Nature. 2011). It might be due to uncontrolled or suboptimal ubiquitination reactions associated with purified proteins. Nevertheless, we have now provided blots with shorter exposure, which hopefully would look better in the revised manuscript. The blot that showed from 35 kDa was provided in the Supplementary Fig. 7, as there was interfering background staining from the antibody heavy chains.